# Consideration of Kinase Inhibitors for the Treatment of Hydrocephalus

**DOI:** 10.3390/ijms24076673

**Published:** 2023-04-03

**Authors:** Bonnie L. Blazer-Yost

**Affiliations:** Biology Department, Indiana University—Purdue University, 723 West Michigan Street, Indianapolis, IN 46202, USA; bblazer@iu.edu

**Keywords:** choroid plexus epithelia, cerebrospinal fluid, transepithelial epithelial ion transport, transient receptor potential vanilloid 4 (TRPV4), sodium, potassium, 2 chloride transporter (NKCC)

## Abstract

Hydrocephalus is a devastating condition characterized by excess cerebrospinal fluid (CSF) in the brain. Currently, the only effective treatment is surgical intervention, usually involving shunt placement, a procedure prone to malfunction, blockage, and infection that requires additional, often repetitive, surgeries. There are no long-term pharmaceutical treatments for hydrocephalus. To initiate an intelligent drug design, it is necessary to understand the biochemical changes underlying the pathology of this chronic condition. One potential commonality in the various forms of hydrocephalus is an imbalance in fluid–electrolyte homeostasis. The choroid plexus, a complex tissue found in the brain ventricles, is one of the most secretory tissues in the body, producing approximately 500 mL of CSF per day in an adult human. In this manuscript, two key transport proteins of the choroid plexus epithelial cells, transient receptor potential vanilloid 4 and sodium, potassium, 2 chloride co-transporter 1, will be considered. Both appear to play key roles in CSF production, and their inhibition or genetic manipulation has been shown to affect CSF volume. As with most transporters, these proteins are regulated by kinases. Therefore, specific kinase inhibitors are also potential targets for the development of pharmaceuticals to treat hydrocephalus.

## 1. Hydrocephalus

Hydrocephalus is a devastating condition that is considered a spectrum of diseases all characterized by excess cerebrospinal fluid (CSF) in the brain. There are several methods of classification within this spectrum. A functional classification is based on whether the cause of the hydrocephalus results in an obstruction to the flow of CSF within the ventricles (e.g., aqueductal stenosis or tumors), resulting in “noncommunicating” hydrocephalus, or whether there is an unobstructed ventricular flow, termed “communicating” hydrocephalus. The latter can be caused by either the overproduction or underabsorption of CSF. Alternatively, hydrocephalus can be classified as “secondary” (acquired) due to an exogenous precipitating cause or “primary” (congenital) [1]. It has been estimated that as much as 40% of primary hydrocephalus has a genetic component encompassing a wide variety of defects [2,3,4,5,6]. Secondary hydrocephalus, the more commonly occurring form of the disease, can be a consequence of intracranial hemorrhage, infection, trauma, or tumors. Additionally, there are idiopathic forms of hydrocephalus, particularly prevalent in the elderly, where it is unclear whether the disease has a genetic or acquired basis. Given this complexity, Rekate suggested a global definition: “Hydrocephalus is an active distention of the ventricular system of the brain resulting from inadequate passage of cerebrospinal fluid from its point of production within the cerebral ventricles to its point of absorption into the systemic circulation”. This author also suggested that the field could benefit from a better method of classifying hydrocephalus [7].

The most visible, and therefore the most well-known, form of hydrocephalus occurs in children where the patients’ heads are enlarged because the skull is not ossified and the buildup of CSF causes a cranial distension. Pediatric hydrocephalus is observed in approximately 1 in 1000 live births in high-income countries, such as the U.S., where it is due primarily to intraventricular hemorrhage in premature infants [8,9]. The incidence of pediatric hydrocephalus is substantially higher in low-income countries, primarily due to the increased incidence of postinfectious hydrocephalus [10]. Despite the visibility of pediatric hydrocephalus, an international meta-analysis that was stratified by age showed that the incidence of hydrocephalus in the elderly is approximately twice that of the pediatric population [11].

Posthemorrhagic hydrocephalus (PHH) is not just a disease of premature infants. PHH can occur at any age, especially in the older population, who are more prone to strokes. Post-traumatic hydrocephalus (PTH) is a secondary consequence of traumatic brain injury (TBI). The Centers for Disease Control reports approximately 1.5 million cases of TBI annually in the United States. The reported prevalence of PTH varies considerably, but some estimates suggest that as high as 80% of TBI patients deal with post-traumatic hydrocephalus of varying severity. Estimates specific to current and former military service personnel report some degree of hydrocephalus in as many as two-thirds of members who suffered moderate to severe TBI [12,13,14,15].

Long-standing overt ventriculomegaly in adults (LOVA) is a heterogenous disease thought to arise in childhood, followed by an asymptomatic, possibly compensated, phase and then an adult-onset symptomatic phase with a progressive course [16]. The etiology remains speculative, and there is some controversy as to the most appropriate surgical intervention [17,18].

Elderly patients may suffer from a poorly understood and underdiagnosed disease called “idiopathic normal pressure hydrocephalus” (iNPH). Often misdiagnosed as Parkinson’s, this age-related ventriculomegaly is characterized by urinary incontinence, gait instability, and cognitive impairment that progresses to dementia if not treated. This condition has been estimated to occur in up to 6% of patients over 80 and, if surgically treated, is one of the few reversible causes of cognitive impairment in the elderly [19,20,21,22].

Currently, the only effective treatment for any form of hydrocephalus is surgical intervention. The most common surgery is the placement of a shunt to redirect CSF from the brain to another body cavity for reabsorption. Shunts are prone to malfunction and infection and require additional, often repetitive, surgeries due to blockage, mechanical malfunction, or infection [23,24,25,26,27]. For example, in the pediatric population, 50% of shunts fail within the first 2 years [28]. Shunts also dictate the constant need for timely access to a hospital that provides neurosurgical care, and each shunt revision is associated with a higher risk of neurological impairment [29,30,31].

Endoscopic third ventriculostomy (ETV) is a surgical alternative to shunting in some patients, such as those with noncommunicating hydrocephalus. This procedure involves the creation of a small opening in the third ventricle toward the subarachnoid space to create a bypass for CSF circulation from the ventricles to the subarachnoid space. A third type of intervention is ETV in conjunction with irreversible cauterization of the choroid plexus in the lateral ventricles in order to limit the amount of CSF produced [24,25,26].

There are no long-term pharmaceutical treatments for hydrocephalus, and the potential candidates that have entered clinical trials were either ineffective or fraught with unacceptable side effects [1]. To initiate an intelligent design for the treatment of hydrocephalus, it is necessary to understand the biochemical changes underlying the pathology of this chronic condition. Hydrocephalus-associated sequelae, such as headache, papilledema, vision and sleep disturbances, gait abnormalities, and cognitive changes, are shared in pediatric hydrocephalus, PTH, iNPH, and PHH. The overlap in symptoms, as well as the efficacy of shunt insertion, suggests that the various disease etiologies share common underlying cellular and molecular mechanisms.

One potential commonality in the various forms of hydrocephalus is an imbalance in fluid–electrolyte homeostasis. This can be due to the overproduction or underabsorption of CSF. Alternatively, there may be a change in the ionic composition of the CSF, leading to fluid accumulation via osmotic forces. Regardless of the initiating cause, treating the changes in fluid–electrolyte balance may provide an effective treatment. Therefore, good targets for drug development are the electrolyte and water transport proteins of the choroid plexus epithelium (CPe) that are responsible for the production and unique composition of the CSF. Given the dynamic changes in the characteristics of the CSF according to circadian rhythms and the need for immediate homeostatic changes in normal physiological responses, key transporters are regulated on an acute basis. This type of regulation most typically involves intracellular kinase-mediated signaling pathways.

## 2. CSF Production

The CP is a complex tissue found in the brain ventricles. This small structure produces the majority of the CSF and is one of the most secretory tissues in the body, producing approximately 500 mL of CSF per day in an adult human [32]. The CP is made up of a fenestrated capillary bed surrounded by choroid plexus epithelial cells (CPe), which form the blood–CSF barrier (Figure 1). The CSF production is controlled by specific electrolyte and water channels and transporters found in the CPe. To form the CSF, selected components of a plasma filtrate characteristic of fenestrated capillaries are transported across the barrier epithelial cells [32,33]. However, the composition of the CSF indicates that it is not simply a plasma filtrate. In addition, the CSF composition changes on a normal diurnal cycle as well as in response to pathophysiological changes, such as trauma and infection, exemplifying the remarkably dynamic nature of this process and indicating the complex regulation of the CSF composition by the CPe cells. While many of the CPe transporter proteins are known, the exact nature of total electrolyte and fluid flux across the epithelial cells remains incompletely characterized. The identity of the intracellular regulatory proteins and biochemical pathways that control these important transport proteins remains largely undefined.

In this manuscript two transport proteins, transient receptor potential vanilloid 4 (TRPV4) and sodium, potassium, 2 chloride co-transporter 1 (NKCC1), will be considered. Both have been postulated to play key roles in CSF production, and their inhibition or genetic manipulation has been shown to affect CSF volume. As with most transporters, these important proteins are regulated by kinases. Therefore, specific kinase inhibitors are also potential targets for the development of pharmaceuticals to treat hydrocephalus.

## 3. TRPV4

TRPV4 is a nonselective cation channel that is found in epithelial, endothelial, neural, and glial cells. The channel can be activated by changes in osmotic balance/cell volume, temperature, mechanical stress (pressure), and inflammatory mediators, such as arachidonic acid metabolites [34,35,36,37,38,39,40], causing it to be referred to as a hub protein, which can integrate multiple stimuli. With regard to the production of CSF, it is in both native CP and CP-derived cell lines [41,42,43,44,45]. When activated, TRPV4 transports Ca^2+^ and Na^+^ into cells, which, in turn, stimulates intracellular Ca^2+^ signaling processes, including the activation of Ca^2+^-sensitive channels (Figure 2). TRPV4 activation also changes the transmembrane Na^+^ gradient, thereby altering electrochemical driving forces and secretory mechanisms. These primary and secondary effects of TRPV4 activation have the potential to alter CSF production by the CPe and modulate fluid/electrolyte balance in other brain regions [42,43,44,45]. In addition to these direct changes, TRPV4 has been shown to be involved in cytokine release by epithelial cells during an NF-kB-mediated inflammatory response in intestinal and respiratory tissues [38,39]. Interestingly, we and others have shown that the activation of TRPV4 causes a substantial decrease in the barrier function/permeability of epithelial cells [40,42,43,44,45], which has implications for changes in immune cell migration and nonspecific ion and small molecule fluxes.

Human genetic diseases involving TRPV4 have been identified, particularly as skeletal dysplasias and arthropathies (Ref. [46], review). Interestingly, the most serious of these are due to gain-of-function mutations. Additionally, the activation of the channel with TRPV4 agonists results in acute effects, most notably pulmonary edema and cardiovascular collapse in several animal models due to increases in epithelial and endothelial permeability [47]. Conversely, the inhibition of TRPV4 by genetic knockout in animals [41,48] or pharmacologically in humans [49] does not substantially alter whole-body homeostasis. The TRPV4 knockout animals produced by two separate groups both had mild phenotypes, which included impaired responses to noxious stimuli [48] and some osmotic sensitivities [41].

Thus, it appears from a whole-body perspective that the pharmacological activation of TRPV4 is physiologically deleterious, while the inhibition of the channel activity is relatively innocuous. These findings support the contention that TRPV4 is activated as a stress response. This role is enhanced by the ability of TRPV4 to act as a hub protein capable of integrating exogenous mechanical and chemical signals.

In a genetic rat model of hydrocephalus, the *Tmem67* model, treatment with TRPV4 antagonists ameliorated hydrocephalic development [50]. In continuous choroid plexus epithelial cell lines of both porcine and human origin, TRPV4 agonists stimulate both electrolyte and water flux across the epithelia, and this movement is blocked by two structurally dissimilar TRPV4 antagonists [43,44,45]. These in vivo and in vitro data suggest that targeting TRPV4 activity either directly or indirectly through intracellular TRPV4 regulators may be a viable strategy for the treatment of hydrocephalus.

TRPV4 is regulated by an array of intracellular signaling mechanisms, many of which are poorly characterized and vary in a tissue-specific manner. The channel contains six membrane-spanning domains with both the N- and C-termini on the cytoplasmic side of the membrane and, therefore, is accessible to a range of intracellular kinases and other regulators, including ubiquitination. Kinases, which have been shown to regulate TRPV4 activity, include serum glucocorticoid induced kinase 1 (SGK1), protein kinase A (PKA), protein kinase C (PKC), Src and with no lysine (WNK) kinases 1 and 4. The data are briefly summarized below, but for a more detailed insight, the reader is referred to two recent reviews [46,51].

SGK1 has been shown to phosphorylate TRPV4 on serine 824, which lies in a calmodulin binding domain that is important for interaction with F-actin. The phosphorylation of this site modulates membrane expression as well as agonist sensitivity and Ca^2+^ influx [52,53,54]. PKA also phosphorylates serine 824 and enhances membrane trafficking [55,56,57]. WNKs increase the membrane expression of TRPV4 but may do so via a secondary effect [58].

PKC phosphorylates the residues serine 165, tyrosine 175, and serine 189 and thereby enhances the sensitivity of TRPV4 in the membranes of cultured renal and dorsal root ganglion cells [55,56]. Src phosphorylates the residues tyrosine 110 and tyrosine 805, thereby sensitizing TRPV4 to heat, shear stress, and hypotonic swelling [59]. As reviewed by Hochstetler et al. [51], G-protein-coupled receptors in several tissues also modulate TRPV4 activity via PKA, Src, and PKCa.

As was pointed out by Darby and colleagues [46], when contemplating drug treatments, it may be possible to elicit a more tissue-specific effect by targeting one of the intracellular signaling cascades that regulates TRPV4 and, thereby, minimize systemic side effects. The results summarized above represent an amalgamation from many different cell types. Recently, a continuous human CPe cell line, the HIBCPP line, having a phenotype consistent with the in vivo CPe, was characterized and used to study the effects of activation and/or inhibition of several kinases on agonist-stimulated TRPV4 activity [45]. Channel activity specific to CPe was measured as transepithelial electrolyte flux stimulated in response to a TRPV4 agonist. The channel activity measurements were performed using Ussing-style electrophysiology, which monitors transepithelial electrogenic ion flux using a noninvasive technique [60,61]. It should be noted that electroneutral transepithelial transport is not detected using this method.

In the HIBCPP cell line, the TRPV4 agonist, GSK1016790A, was shown to stimulate a multiphasic transepithelial electrogenic ion flux that was accompanied by fluid secretion. This secretory movement was not solely a function of TRPV4-mediated Ca^2+^ influx because it was not mimicked by treatment with ionomycin, an ionophore that allows Ca^2+^ influx across the plasma membrane, or by thapsigargin, which blocks Ca^2+^ uptake into the endoplasmic reticulum. In fact, the increase in intracellular Ca^2+^ in both instances partially or completely blocked the TRPV4 transepithelial electrolyte flux. Likewise, the effects of inhibitors and activators of phospholipase C were consistent with the finding that increases in intracellular Ca^2+^ decrease TRPV4 activation [45].

Pretreatment of the HIBCPP cells with either of two PKC inhibitors, tamoxifen or Go6976, inhibited a subsequent response to TRPV4. Interestingly, treatment of the cultures with phorbol 12,13-dibutyrate, an activator of PKC, stimulated a transepithelial electrogenic ion flux that was blocked by a TRPV4 antagonist. These data indicate that the TRPV4 activation of PKC is important for the regulation of transepithelial transport in the CPe cells [45].

Conversely, treatment of the HIBCPP cells with effectors of PKA had no effect on either baseline or TRPV4-stimulated electrogenic transepithelial ion flux (Blazer-Yost, et al., unpublished data). Effectors of other kinases, such as Src and SGK1, which have been shown to modulate TRPV4 activity in other tissues, have not yet been reported in the choroid plexus literature.

## 4. NKCC1

NKCC1 is another transport protein that has emerged as a potential key player in the process of CSF production. NKCCs were first described in 1980 [62], and over a decade later, two isoforms were cloned [63,64,65]. NKCC2 is renal specific [66], while NKCC1 is ubiquitously expressed and is involved in the maintenance of cell volume in many tissues. Like TRPV4, NKCC1 is expressed on the apical membrane of the CPe [67].

A somewhat puzzling question is how the CPe can secrete large amounts of fluid into the ventricular space in the absence of a substantial osmotic gradient. Zeuthen and colleagues proposed that several electrolyte cotransporters, including NKCC1, also transport substantial amounts of water irrespective of the osmotic gradient [68,69]. However, recent single-particle cryo-electron microscopy, combined with molecular dynamic simulation, showed that while water can move passively through human NKCC1, the number of water molecules is small, ruling out the “water-pump” hypothesis of Zeuthen [70]. Currently, one has to conclude that the processes that result in the production of half a liter of CSF per day in an adult human remain a matter of speculation.

However, the importance of NKCC1 to CSF production, particularly under stress conditions, is well established. Thirty years ago, experiments in dogs showed that ventriculo-cisternal perfusion with bumetanide, a relatively specific inhibitor of NKCC, decreased CSF production by approximately 50% [71]. More recently, the group of Kahle infused bumetanide into rat ventricles to show that this inhibitor caused a decrease in CSF production after the induction of mild posthemorrhagic hydrocephalus [72]. There is agreement in the literature that systemic routes of bumetanide delivery are not effective [72,73,74], presumably due to the poor permeation of the drug across the cell membrane and the location of NKCC1 on the apical membrane of CPe, facing the CSF.

It is unclear from the recent studies whether the activity of NKCC1 is crucial for CSF production under normal physiological conditions or is only effective in stress situations when the CPe is stimulated to oversecrete fluid. In normal mice, one study specifically reported that intraventricular treatment with bumetanide decreased CSF production by approximately half [75]. Karimy et al. did not specifically address the effect of bumetanide under normal conditions, but extrapolation of data from different experiments suggests a bumetanide-induced decrease in ventricular volume in animals that were not subjected to conditions, modeling posthemorrhagic hydrocephalus [72]. Both of these studies suggest the importance of NKCC1 under normal conditions. Conversely. Bothwell et al. found no effect of NKCC1 antagonism on CSF production in healthy rats [74]. Additionally, NKCC1 knockout mice have no apparent change in brain ventricular size, although MRIs have not yet been performed in this model [76].

Another controversial issue with regard to the role of NKCC1 in CSF production is the direction of the transport of the three electrolytes and, potentially, water (Figure 3). In most tissues, NKCC1 has been shown to function in the influx mode, which transports the three electrolytes into the cell as part of the cell volume regulatory mechanism and functions to normalize intracellular Cl^−^. In the 1990s, Keep and colleagues suggested that NKCC1 contributed to electrolyte and water secretion into the CSF by working in a “reverse” mode, transporting these ions out of, rather than into, the CPe and, thus, contributing to CSF production [77,78]. However, another group indicated that NKCC1 was running in its more traditional mode, and the influx of ions was not only constitutive but important for establishing and maintaining the gradient-driven uptake of K^+^, resulting in a K^+^ reabsorption from the CSF. Using an isolated CPe preparation, their results suggest that physiologically relevant increases in extracellular K^+^ stimulate the cotransporter, resulting in rapid K^+^ uptake [79].

Recently, the Lehtinen group showed that in mice, the CSF K^+^ concentration decreased by about half from the time of birth to day 7. This was paralleled by an increase in the NKCC1 protein, leading this group to conclude that NKCC1 working in the influx mode was crucial for K^+^ clearance from the CSF immediately after birth. Building on this finding, they showed that the adenoviral-mediated overexpression of NKCC1 in the CPe reduced ventricle size in a mouse model of obstructive hydrocephalus [80].

Thus, despite a substantial number of studies, there seems to be little agreement as to whether the transporter is contributing to an inward flux of Cl^−^ necessary to maintain the intracellular concentration for the maintenance of cell volume and Cl^−^ secretion and/or K^+^ absorption by the CPe [81] or whether it is the major transporter for Na^+^, Cl^−^ and water secretion by the CPe [75]. This question can be simplified to the question of whether NKCC1 is acting in an influx direction as it does in most other cells, or it is transporting electrolytes and water in an efflux direction. For more detail regarding the on-going controversy, the reader is referred to an interesting CrossTalk series in the *Journal of Physiology* [82,83,84,85]. As suggested by multiple authors within the CrossTalk series and associated comments, NKCC1 may be poised close to equilibrium with the ability to change direction as required to maintain cellular volume/electrolyte composition.

The major kinases that regulate NKCC are the WNK (with no lysine) and SPAK/OSR1 (SPS1-related proline/alanine-rich kinase/oxidative stress-responsive kinase 1) pair. WNK is sensitive to changes in intracellular Cl^−^, which is determined, in part, by NKCC activity. WNK phosphorylates and activates SPAK, which, in turn, phosphorylates and activates NKCC. Interestingly, this pair also activates Na^+^-Cl^−^ transporters (NCCs), which transport Na^+^ and Cl^−^ into cells. In addition, these kinases phosphorylate and inhibit K^+^-Cl^−^ transporters (KCCs), which extrude Cl^−^ and K^+^ from cells (Refs. [86,87], reviews). SPAK was the most abundantly expressed kinase in rat choroid plexus by the technique of RNA-seq [88].

In an 8-week-old rat model of intraventricular hemorrhage, Karimy et al. demonstrated enhancement of the phosphorylated levels of both NKCC1 and SPAK with no corresponding increases in total protein, indicating the activation of both proteins. In the rat model, the increased production of CSF was inhibited by intracerebroventricular bumetanide and inhibitors of SPAK activity [72].

## 5. Overlap in the Biochemical Pathways and Kinases

It is likely that TRPV4 and NKCC1 are both important for the maintenance of CSF volume and composition particularly under stress conditions imposed by disease or injury. The activation of TRPV4 is a stress response due to changes in osmotic balance, pressure, or inflammatory mediators, while the activation of NKCC1 is more specific and is due to changes in the concentration of defined electrolytes. In concert with the multiple inputs that activate or sensitize TRPV4, this nonselective cation channel appears to be regulated by multiple kinases (Figure 3). On the other hand, NKCC1, predominately regulated by the WNK-SPAK/OSR1 kinase pair, may be poised close to equilibrium and capable of transporting K^+^, Cl^−^, and Na^+^ into or out of the CPe as necessary to maintain cell volume homeostasis and/or CSF composition (Figure 3). There may be a degree of overlap in the pathways regulating these two important transporters. For example, in ex vivo studies using isolated choroid plexuses, the SPAK inhibitor closantel and the WNK inhibitor WNK463 had no effect on basal ion flux through NKCC but reduced TRPV4-activated flux (88). Potential areas of cross regulation of kinases have been described in a variety of other tissues. A few of these are summarized below.

Changes in CSF K^+^ concentrations are regulated and change significantly during early postnatal development [80]. In addition, increases in brain K^+^ are also associated with multiple types of brain injury. In renal distal tubule cells, high extracellular K^+^ stimulates an increase in intracellular Cl^−^, which, in turn, causes a conformational change in WNK1, inhibiting its kinase activity but promoting a scaffolding complex that results in the phosphorylation and activation of SGK1 [89]. Projecting such changes to a hypothetical CPe pathway would mean that the increase in extracellular K^+^ could ultimately result in the activation of TRPV4 via SGK1 and a decrease in the activation of NKCC and NCC via an inhibition of WNK-SPAK phosphorylation. This may also result in a stimulation (removal of inhibition) of KCC activity, leading to an increased K^+^ reabsorption via KCC that is found on the basolateral membranes of CPe.

TMEM16A and cystic fibrosis transmembrane conductance regulator (CFTR) are two Cl^−^ channels found in the CPe. TMEM16A is a Ca^2+^-sensitive channel that can be activated by TRPV4-stimulated Ca^2+^ influx, resulting in the secretion of Cl^−^ into the CSF [90,91,92]. In primary cultures of CPe, TRPV4 has been shown to physically interact and stimulate TMEM16A, and this has been postulated to regulate aquaporin water channels, resulting in the secretion of both water and Cl^−^ into the CSF [93]. Somewhat surprisingly, TRPV4 has recently been shown to activate CFTR [94]. The exact mechanism for the latter activation has not been elucidated but would, presumably, also lead to Cl^−^ secretion into the CSF. The change in intracellular Cl^−^ concentration can, theoretically, also link the TRPV4 and NKCC1 pathways. The WNK family of kinases are bona fide Cl^−^ sensors within the physiological range. Cl^−^ binding inhibits autophosphorylation and activation [95,96,97,98]. Cl^−^ secretion and the resulting decrease in intracellular concentration could lead to the activation of WNK, followed by the phosphorylation and activation of SPAK/OSR1 and, subsequently, NKCC1.

In a human salivary gland cell line, increases in intracellular Ca^2+^ that were stimulated in response to hypotonic stress were paralleled by an increased phosphorylation of OSR1. The phosphorylation was blocked by Ca^2+^ channel blockers and a Ca^2+^ chelator. The authors suggested that Ca^2+^ upregulated the WNK-SPAK/OSR1/NKCC pathway, and they hypothesized that TRPV4 may be a component of the pathway (Figure 3) [99].

The signaling elements listed above are not exhaustive but serve to indicate that the kinase-regulated pathways controlling TRPV4 and NKCC1 have multiple points of potential interaction. It is not unexpected that a tissue like the CPe will have a complex regulatory network in order to maintain secretory function in the face of a dynamic extracellular milieu.

## 6. Conclusions

Hydrocephalus in a multispectrum disease that strikes people of all ages. Although there are multiple causes, most forms of the disease are treated with the same surgical interventions. The efficacy of surgical intervention, combined with similarities in the symptomatic presentation in the multiple disease entities, suggests that the biochemical pathways underlying the pathology may be similar and may result, in part, from electrolyte and fluid imbalances. Preclinical studies have shown that systemic treatment with TRPV4 antagonists [50] or intraventricular infusion of an NKCC1 or SPAK inhibitors [72,100] decreases hydrocephalic severity. The regulation of both transport proteins involves kinases that are potential targets for the development of pharmaceutical interventions to treat multiple forms of hydrocephalus.

## Figures and Tables

**Figure 1 ijms-24-06673-f001:**
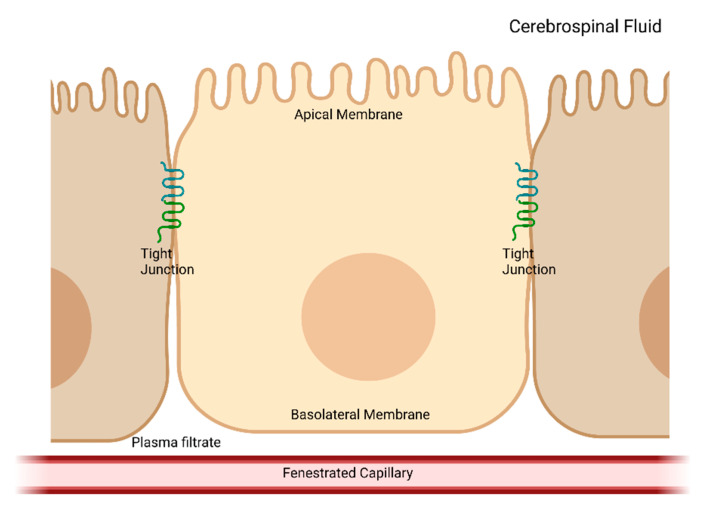
Schematic of choroid plexus epithelial cells. The choroid plexus epithelial cells form a cellular monolayer wherein the cells are connected by tight junctions to form a barrier epithelium separating the plasma filtrate formed by a fenestrated capillary network from the cerebrospinal fluid. In the intact epithelium, the cells are polarized with the apical membrane above the junctional complex and the basolateral membrane facing the plasma filtrate. There are distinct transport proteins in the apical and basolateral membranes (not shown) that contribute to the formation of the cerebrospinal fluid.

**Figure 2 ijms-24-06673-f002:**
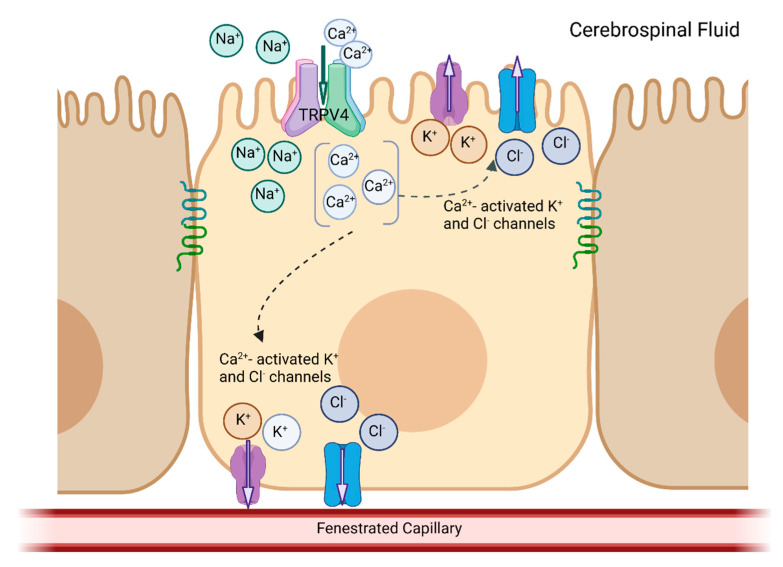
Transient receptor potential, vanilloid 4 (TRPV4) in the apical membrane of choroid plexus epithelial cells. The activation of TRPV4, located on the apical membrane, results in the influx of calcium (Ca^2+^) and sodium (Na^+^) into the cells. The increased Ca^2+^ secondarily activates Ca^2+^-sensitive channels, resulting in net electrolyte flux across the epithelium.

**Figure 3 ijms-24-06673-f003:**
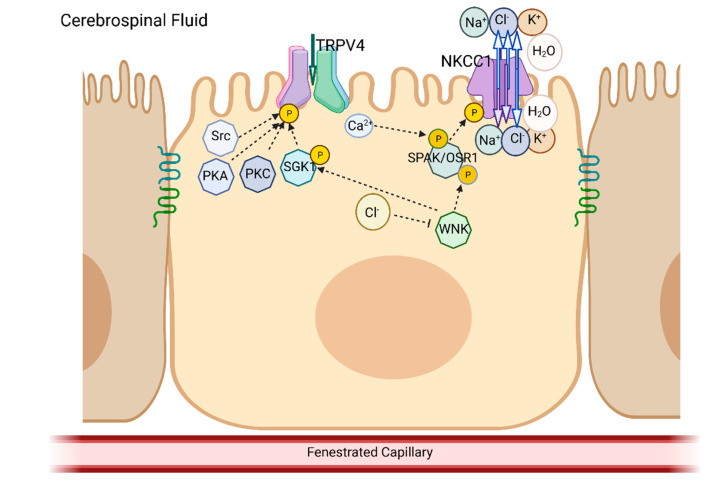
Schematic of the kinases involved in the activation of TRPV4 and NKCC1. As shown, multiple kinases have been shown to phosphorylate TRPV4, while the SPAK/OSR1 kinase complex appears to be the main activator of NKCC1. Potential interactions between the kinases, which control the activity of these two transport proteins, are shown in the diagram. Please see the main text for additional details. TRPV4, transient receptor potential vanilloid 4; NKCC1, sodium, potassium, 2 chloride cotransporter 1; SGK1, serum glucocorticoid induced kinase 1; PKA, protein kinase A; PKC, protein kinase C; WNK, with no lysine kinase; SPAK/OSR1, SPS1-related proline/alanine-rich kinase/oxidative stress-responsive kinase 1.

## Data Availability

There are no new data in this review article.

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
