# Peer review of "Consideration of Kinase Inhibitors for the Treatment of Hydrocephalus"

_ijms, 2023, doi:10.3390/ijms24076673_

Round 1

Reviewer 1 Report

In this review manuscript, Blazer-Yost provide a thoughtful summary of current possible targets for pharmacological treatment of hydrocephalus, particularly with kinase inhibitor and the role of TRP4 and NKCC on CSF physiology.

It is a well written manuscript with updated information.
I would only suggest minor changes in the Introduction, specially with regards to clinical aspects of hydrocephalus:

1. The definition of hydrocephalus can be improved. I don't think that the colloquial "water-on-the-brain" is appropriate to the future readers/audience of the paper. Instead, the Rekate's definition suits better and could be stated, referred and explored in the text: "active distension of the ventricular system..." (see: Rekate HL. The definition and classification of hydrocephalus: a personal recommendation to stimulate debate. Cerebrospinal Fluid Res. 2008 Jan 22;5:2. )

2. On referring to "noncommunicating" hydrocephalus (obstructive), it is not precise to give only the example of acqueductal stenosis. There are many other causes of obstruction within the ventricular system that deserve to me mentioned, such as brain tumors. Besides, many authors argue that every hydrocephalus has an obstruction component - either within the ventricles (obstructive, non-communicating) or in the subarachnoid space (communicating). I recognize that it is controversial. However, I would recommend the author to change the text (lines 31-34). If the obstruction in within the ventricular system, non-communicating. If the flow is not impaired IN THE VENTRICLES, communicating.

3. Regarding data on the epidemiology of hydrocephalus, it is important to cite and present data from Isaacs et al: Isaacs AM, Riva-Cambrin J, Yavin D, Hockley A, Pringsheim TM, Jette N, Lethebe BC, Lowerison M, Dronyk J, Hamilton MG. Age-specific global epidemiology of hydrocephalus: Systematic review, metanalysis and global birth surveillance. PLoS One. 2018 Oct 1;13(10):e0204926..

Even though this article was close to the cited one (Dewan et al, 2018), they have complementary information and it is a very important reference in the field.

4. On the description of VP-shunt malfunctions, it would be welcome to refer that each episode of VPS revision is associated with higher risk of neurological impairment. 

5. The explanation of endoscopic third-ventriculostomy can be improved with the information that the "small opening" of the third ventricle is made towards the subarachnoid space, creating a bypass for CSF circulation from the ventricles to the subarachnoid space.

Author Response

I thank the reviewer for their comments and the helpful suggestion for improvement of the clinical description and, in some cases, more precise wording.  I have made all the suggested text changes and added all of the suggested references.

Reviewer 2 Report

Interesting topic, but paper needs some revisions. Look carefully at these points:

- Lines 8-24. Abstract. "The choroid plexus, a complex tissue found in the brain ventricles, is one of the most secretory tissues in the body, producing approxi mately 500 ml of CSF per day in an adult human" Some sentences like this one are very obvious and they should be removed.

- Lines 90-93: "Therefore, good targets for drug development are the electrolyte and water transport proteins of the choroid plexus (CP) that are responsible for production and unique composition of the CSF." It is not clear what is the aim of this paper and what this paper add new to the literature. Please revise this part.

- Lines 81-83: "Hydrocephalus-associated sequelae such as headache, vision and sleep disturbances, gait abnormalities and cognitive changes are shared in pediatric hydrocephalus, PTH, iNPH, and PHH. " But also papilledema among signs and symptoms. Consider this reference:  PMCID: PMC8879089  --  DOI: 10.3390/medicina58020281

- Lines 34-36: "The latter can be caused by either overproduction or under absorption of CSF. Alternatively, hydrocephalus can be classified as “secondary” (acquired) due to an exogenous precipitating cause or “primary” (congenital) [1]. " Among hydrocephalus, authors should consider also LOVA. Look at these 2 important refs: -- PMCID: PMC8872207  -- DOI: 10.3390/ijerph19041926 --  DOI: 10.1007/s00701-021-04983-0

- What does this paper add new to the literature about TRPV4 ? This point needs to be addressed better

- "Traditionally, pharmacological interventions or CSF drainage have been used to reduce ICP elevation due to over production of CSF. However, these drugs are used only as a temporary solution due to their undesirable side effects" Consider: PMCID: PMC6456952 -- DOI: 10.1186/s12987-019-0129-6

- What about the role of acetazolamide in relation to the TRPV4 and NKCC1 proteins? Bothwell SW et al. reproted that acute antagonism of NKCC1 and TRPV4 proteins at the choroid plexus does not reduce CSF secretion in healthy rats.

- Lines 111-115. Figure 1 is similar to figure 2. Eventually merge them.

- Lines 363: "The examples listed above are not exhaustive but serve to indicate that the kinase... " What examples? specify them.

- Lines 374-379: "Regulation of both transport proteins involve kinases which are potential targets for pharmaceutical interventionsThis is a very short conclusion. Please improve it. What does this review want to highlight?

Round 2

Reviewer 2 Report

Authors solved all my criticisms.